# The Influence of Walking Limitations on Daily Life: A Mixed-Methods Study of 14 Persons with Late Effects of Polio

**DOI:** 10.3390/ijerph19138157

**Published:** 2022-07-03

**Authors:** Christina Brogårdh, Jan Lexell, Catharina Sjödahl Hammarlund

**Affiliations:** 1Department of Health Sciences, Lund University, S-221 00 Lund, Sweden; jan.lexell@med.lu.se (J.L.); catharina.sjodahl_hammarlund@med.lu.se (C.S.H.); 2Department of Neurology, Rehabilitation Medicine, Memory Disorders and Geriatrics, Skåne University Hospital, S-221 85 Lund, Sweden; 3The PRO-CARE Group, School of Health and Society, Kristianstad University, S-291 39 Kristianstad, Sweden

**Keywords:** activities of daily living, disabled persons, post poliomyelitis syndrome, qualitative research, rehabilitation, walking limitations

## Abstract

Reduced walking ability is common in persons with late effects of polio (LEoP). Here, we explored how many walking limitations persons with LEoP perceive, and how these limitations influence daily life, by using a mixed-methods design. Fourteen persons (mean age 70 years, whereof 7 women) with LEoP responded to the Walking Impact Scale (Walk-12), and were interviewed individually. Qualitative data were analysed by systematic text condensation, and each quotation was deductively analysed in relation to the items in Walk-12. Running was perceived as most limited, whereas walking indoors without using support was perceived as least limited. A majority (>70%) were moderately to extremely limited in standing or walking, in walking speed and distance, which affected concentration and effort, as well as gait quality aspects. The limited walking ability intruded on many everyday activities, both indoors and outdoors, which affected social participation negatively. To increase safety when walking and reduce the fall risk, various strategies were adopted such as using aids, walking carefully, and avoiding risky activities. In conclusion, LEoP-related walking limitations influence participants’ activity and participation greatly. By using both the Walk-12 scale and face-to-face interviews, an increased understanding of how walking limitations influence everyday life was achieved.

## 1. Introduction

Late effects of polio (LEoP) is a chronic progressive neuromuscular condition that can arise decades after the initial poliomyelitis infection [1]. The underlying causes of LEoP are not entirely clear, but it is generally agreed that new or increased impairments occur due to a distal degeneration of axons in the enlarged motor units that developed during the recovery of the acute paralytic polio phase. At the age of around 50 years, when the “normal” age-related loss of motor neurons occur, many people with LEoP experience increased muscle weakness in the lower limbs, muscle fatigability, general fatigue and musculoskeletal pain [2]. The increased impairments can lead to reduced balance [3,4] and walking limitations in daily life [5,6]. The walking limitations can, in turn, lead to physical inactivity [7], falls and fear of falling [5] as well as restrictions in perceived participation [8]. Thus, to increase safety whilst walking is therefore an important goal in rehabilitation for people with LEoP.

One rating scale that can be used to assess self-perceived walking limitations in daily life is the Walking Impact Scale (Walk-12) [9]. Walk-12 is based on the Multiple Sclerosis Walking Scale (MSWS-12) [10] and consists of 12 items that aim to explore aspects of self-perceived walking limitations in activities related to standing and walking. It is shown to be a useful rating scale in persons with various neurological conditions [9,11,12,13,14], including LEoP [15,16]. Walk-12 is easy to administer and reflects a broader perspective of walking limitations than objective gait performance tests [15]. It has also been shown to have sufficient measurement properties in people with LEoP [17]. However, one disadvantage with the Walk-12 is that it includes predefined questions, which may limit a clear understanding of the everyday problems that people with walking limitations following LEoP may perceive. Qualitative interviews can therefore be an important complement to a rating scale [18]. The ratings in Walk-12 provide information about to what extent the participants perceive walking limitations, whereas the interviews provide nuances and detailed understanding of the limitations. Thus, by collecting data from both a rating scale and interviews, an enhanced knowledge of the problems is obtained. To the best of our knowledge, no previous study has combined these two methods to assess self-perceived walking limitations among persons with LEoP. The aim of this study was therefore to explore how much walking limitations people with LEoP perceive, and how these limitations influence daily life.

## 2. Materials and Methods

### 2.1. Research Design

This study applies a mixed-methods design [18]; quantitative data were obtained by the Walk-12 scale and qualitative data by semi-structured face-to-face interviews. The study was part of a larger research project about falls, fear of falling, self-perceived impairments, and walking limitations in persons with LEoP [5]. In the present study, only data about self-perceived walking limitations are reported.

### 2.2. Participants

Participants with a confirmed history of acute poliomyelitis in their youth and with new impairments after a period of functional stability were recruited from a post-polio rehabilitation clinic in southern Sweden. They had previously taken part in the larger project about falls and fear of falling [5]. From that cohort [5], a wide range of variations among the participants were sought by strategically selecting them regarding gender, age, functional level, years with LEoP, and fall frequency. Persons with no walking ability at all were excluded from the study.

Eighteen persons were contacted by telephone and informed about the present study, and fourteen (seven women and seven men) agreed to participate. Their mean age was 70 years, the mean age at the acute poliomyelitis infection was 4 years, and the mean duration of new symptoms of LEoP was 26 years. Most of the participants lived with a partner and had part-time disability pension or old age pension. A majority had moderate to severe disability following LEoP. All of them were able to walk at least 100 m, and most of them used a stick and/or orthotic device during ambulation (see Table 1).

### 2.3. Ethics

All persons received oral and written information prior to inclusion in the study, and gave informed consent to participate. The principles of the Declaration of Helsinki were followed, and the study was approved by the Regional Ethical Review Board, Lund, Sweden (Dnr: 2014/186).

### 2.4. Data Collection

First, the participants rated their self-perceived walking limitations by responding to the Walk-12, which was sent home to them before the interviews. The Walk-12 was developed by Holland et al. [9] and has shown good measurement properties in persons with LEoP [17]. The scale consists of 12 items on self-perceived limitations during the past two weeks regarding walking, running, climbing stairs, standing, balance, walking distance and effort, need for support indoors and outdoors, gait quality aspects and concentration when walking, due to their LEoP. It has five response options ranging from 1 (not at all limited) to 5 (extremely limited). 

The participants brought the Walk-12 ratings with them in connection with the interviews, which took place at a post-polio rehabilitation clinic in southern Sweden. All interviews were performed individually by the first author (CB, physiotherapist) and the last author (CSH, physiotherapist and psychologist). The interviews were based on a semi-structured interview guide that included open-ended questions regarding: experiences and consequences of ageing with LEoP and how the walking limitations influence daily life. Follow-up questions such as: ‘Can you give an example?’ and ‘Please, describe’ were used. Each interview lasted between 60 and 90 min. The interviewers had professional experience of persons with LEoP but no therapeutic relation with the participants. All interviews were audio-recorded and transcribed verbatim. 

### 2.5. Data Analysis

The Walk-12 data were analysed after the interviews, by descriptive statistics. The results of each item and response category are presented as absolute (*n*) and relative numbers (%).

The qualitative data were analysed by systematic text condensation (STC), as it is a well-suited method for cross-case analysis. STC is based on Giorgi´s phenomenological approach to explore the lived experiences of the respondents [19]. The analysis was performed in several steps. First, the interview transcripts were read several times to identify and categorise primary themes related to perceived walking limitations or their impact on everyday life. Meaning units were identified and the content was formulated into codes, which represented the essence of the data. Thereafter, the coded data were organised into subcategories and the general meaning was formulated into aspects representing the content. The subcategories were then organised into categories. Thereafter, each quotation was deductively analysed from its representation with regard to the perceived walking limitations or their impact on everyday life. Each quotation that matched one of the items in Walk-12 received the corresponding number. The analysis was mainly carried out by the first and last authors (CB, CSH), and the findings were validated by the second author (JL, a physician with long experience of persons with LEoP).

## 3. Results

### 3.1. The Ratings of the Items in Walk-12 and the Corresponding Qualitative Data

In Table 2, the ratings of the items in Walk-12 are presented. Almost all participants (85%) reported that their ability to run was moderately to extremely limited. A majority (>70%) also perceived that their LEoP had limited (moderately to extremely) their ability to walk, to maintain balance when standing or walking, their walking distance and walking speed, affected their concentration and effort when walking as well as their ability to walk smoothly. In contrast, approximately one-third of the participants reported that their disability following LEoP had not at all made it necessary for them to use support when walking indoors. 

### 3.2. The Deductive Analysis

The highest representation regarding number of statements in the interviews was about limited balance when standing or walking [46 statements], limited ability to walk [34 statements], and necessary to use support when walking indoors [24 statements]. The result of the deductive analysis is presented below, and follows the order of the items in Walk-12.

#### 3.2.1. Limited Ability to Walk (Item 1)

A majority of the participants (79%) rated that their LEoP had led to limited walking ability. The increased impairments following LEoP, such as muscle weakness in the lower limbs, fatigue and pain in the body, had gradually decreased their walking ability. One man had such problems and described that he could only walk a few steps before back pain appeared. 


*I can only walk three steps, then my back starts hurting…*
 [P8]

The reduced ability to walk affected everyday life greatly. Going shopping, which used to be fun, was now experienced as more difficult and strenuous. To save energy, some of the participants used shopping carts as support. Many also needed a completely flat ground when walking, which affected their ability to engage in various daily activities and leisure activities.


*In many situations, you really are [limited in your walking ability].*
*Especially when it comes to moving around. Like walking in stores, for instance… I don’t like when it´s too crowded, either. When we have been to an ice hockey arena and there are too many people there, there’s immediately a feeling of imbalance, having to parry people and THAT is tough.*
 [P7]

#### 3.2.2. Limited Ability to Run (Item 2)

Eighty-five percent rated that running was moderately to extremely difficult, and they described that it had been so throughout their lives. Many had tried to run when they were younger but said that it was very difficult. For some, running was not possible because the Achilles tendon was too short. The limited ability to run led to problems to perform leisure activities, for example playing football. Gaining increased impairments as a consequence of LEoP made running even harder and sometimes no longer possible, which intruded on social activities and the ability to play with grandchildren. 


*…I tried to run when I was younger, but I can´t do that anymore…*
 [P7]


*When the grandchildren come to visit I play with them, play hide and seek and play with them outside and stuff. But if there’s a lot of running and ball-kicking grandpa is the one who has to do that.*
 [P9]

#### 3.2.3. Limited Ability to Climb Up or Down Stairs (Item 3)

Various degree of limitations when climbing up or down stairs were reported among the participants. Some had minor limitations (29%), while others had more pronounced problems (50%). The participants said that before, when they were younger, climbing stairs did not bother them so much. However, ageing with LEoP meant increased muscle weakness of the lower limbs, especially the most affected one, which reduced movement control and increased their need to support themselves on a banister when climbing up or down stairs. 


*Before I didn’t have to think before climbing down the stairs, but today I like to hold on to banisters if there are any. Before I was able to get down without having to hold on, but I don’t dare to do that anymore. Something has happened to the bad leg, sometimes it’s as if it doesn’t want to cooperate with [me].*
 [P10]

The reduced ability to climb stairs affected the ability to participate in social events, for example going to a dinner party, the theatre or to the ice hockey arena. One participant said that when she was going to see an ice hockey game, she almost always made sure to get a seat where she does not have to climb a lot of stairs. Another person said that he had given up swimming in the sea because he did not know if he would manage the stairs anymore.


*…I used to drive to swim in the sea. I*
*cannot do that longer because you have to go down a stair and I do not know if I am able to climb up again.*
 [P8]

#### 3.2.4. Difficulty to Stand When Doing Things (Item 4)

Many participants (65%) reported that their disability following LEoP had made standing when doing things moderately to extremely more difficult. The reduced ability to stand limited many daily activities, for example taking a shower. For some it was necessary to use a shower board, because of their fear of falling in a standing position. 


*I always sit on it [the shower board] when I shower.*
*Or I stand up at first and then sit down. I’m afraid of standing up in the bathtub even if I’m standing on a rubber mat.*
 [P3]

Standing up and doing things in the kitchen such as peeling potatoes and cooking could also be difficult. Some managed to stand for a while but then had to sit down and rest due to fatigue and pain in the body. To use various aids that they could sit on, for example chairs or bar stools, facilitated the work in the kitchen. After a rest, they could often continue with their work in a standing position again. 


*…I have a stool in the kitchen, a bar stool that we bought when I fell many years ago and couldn’t stand because my foot was sprained again…. I pull it out when I need to, for example when standing and peeling a lot or when standing and working, I can sit sometimes, and that’s great.*
 [P9]

The limited ability to stand also affected participation in social activities. One participant described that the pain in the lower limbs and back increased when standing for a long time.


*…I sing in the choir as well, and we stand quite a lot when there is a concert, and it affects [me]. Because then… it feels both in the legs and back….*
 [P11]

#### 3.2.5. Limited Balance When Standing or Walking (Item 5)

Around two-thirds (65%) reported that their LEoP had limited their balance, moderately or quite a bit, when standing or walking. Too much load on the weaker limb could affect their balance, making it necessary to quickly change position to avoid falls. Some participants also described that when they got up from sitting to standing, they had to stand still for a while to regain balance.


*Nowadays it’s a lot harder to move… If*
*I’m getting something and have to get up. Now I have support and crutches and make sure I stand properly before I take a step.*
 [P5]

Keeping the balance when walking could also be challenging, especially when standing on the weaker leg. Being able to maintain balance when it was dark, indoors as well as outdoors, was also difficult. One strategy that many used was to stand still for a while before they moved on to regain stability.


*Darkness is not good. I*
*now notice that it is not… I get shaky when I walk when it´s dark. So it’s not that fun to be outdoors in the dark yourself, it’s not. Somehow you´re more imbalanced when it’s dark… You can´t see anything…*
 [P7]

#### 3.2.6. Limited Walking Distance (Item 6)

Most of the participants (71%) rated that their LEoP had limited, quite a bit or extremely, how far they could walk. Walking a longer distance often resulted in tiredness of the lower limbs and for some the muscles started aching quite quickly. The muscle fatigue, in turn, led to an increased risk of falling and a fear of falling. Therefore, they could not move around as much as they wanted. As a result, long distances were avoided because they knew they would have problems the day after. Instead, they had learned to listen to their body and walked shorter distances adapted to their ability.


*You walk to the best of your ability… It’s not very far.*
*I walk outside the house, go for a short walk. It’s not very far. I have trouble with walking long distances.*
 [P10]


*… I can´t walk as much nowadays as I would like to… I feel that I´m getting increasingly tired and I get a lot of pain if I, for example, walk as far as I did half a year ago. That really bothers me that I can´t do that anymore…*
 [P7]

#### 3.2.7. Increased Effort When Walking (Item 7)

All participants rated that the LEoP had increased their effort when walking to some extent. Walking up slopes and climbing stairs were particularly strenuous as it required movement control and good muscle strength in the lower limbs. Walking on uneven grounds was also perceived as strenuous.


*…walking on slopes, it’s hard… and*
*the same with stairs… Then you must constantly think about how to place the foot…*
 [P11]

#### 3.2.8. Necessary to Use Support When Walking Indoors (Item 8)

More than half of the participants (57%) reported that they had to use some kind of support when walking indoors, while about one third (36%) did not use any support at all. Those who needed support described that their increased impairments following LEoP had made them more unstable which increased the risk of falling. To increase safety when walking indoors, some had to use walking aids, such as orthoses, a cane, a rollator or a banister. 


*I can’t go to the bathroom without a walker. The risk of falling is too high.*
 [P8]

Another strategy to increase safety when walking was to stand firm and have something to lean on before taking a step. Some also said that even if they did not have to use a walking aid, they unconsciously took support from walls or furniture.


*At home I always walk so I can get support from walls, tables and other similar things. I think I do it unconsciously, if I were to get unstable I can always grab something.*
 [P5]

#### 3.2.9. Necessary to Use Support When Walking Outdoors (Item 9)

Fifty-seven percent reported that their LEoP had made it necessary for them, quite a bit or extremely, to use support when walking outdoors. Several types of walking aids were used among the participants during outdoor walking, for example orthoses, canes, Nordic walkers or rollators. 


*Then I’ve also gotten those dictus bands that help me greatly when I’m outside… I have them on when I know I’ll be walking long distances. I’ve also gotten some kind of orthosis that’s supposed to help stabilize my left leg which is also really good.*
 [P5]

Using different types of walking aids, but also anti-slip soles and good shoes, was especially important during wintertime or when it was slippery outside. Several said that they felt nervous to go out when it was slippery and did not do so without the support of their walking aids. 


*I put studs on my crutch, anti-slips. Then you get to have a pair of sturdy winter shoes. You cannot go out with shoes with plastic soles. You have to make sure that you have the right shoes when it´s slippery. You are also extra careful…*
 [P2]

Choosing the right walking aid in relation to the environment and the ground was important, for example to be able to continue walking in the forest and picking mushrooms.


*It´s difficult to walk with Nordic walkers in forest and land with hilly ground. I have always enjoyed going out into the forest and picking mushrooms but I experience that it has become much more difficult today than before… Then I need support and use my cane. It doesn’t work as well with Nordic walkers in the forest.*
 [P5]

#### 3.2.10. Slowed Down Walking Speed (Item 10)

Most of the participants (71%) reported that they had had to slow down their walking speed, quite a bit or extremely, due to their increased disability following LEoP. Over time, they had learned to take it easy in different situations and not to walk too fast. For example, when they were out walking with friends and family members, they had to keep their own pace and to walk more carefully to reduce the risk of falling and not to physically overload the body. They had also learned to make sure that they always were on time, which was not always easy. The faster they walked, the easier it was to get unstable and fall. 


*You learn that you can’t walk too fast, you need to take it easy. You always have to make sure you’re on time and that’s not always easy. But the faster I walk, the worse it gets. The easier it is to fall, or to feel like you get a bit unstable.*
 [P7]

#### 3.2.11. Ability to Walk Smoothly (Item 11)

Nearly three quarters (71%) rated that their LEoP had affected, quite a bit or extremely, how smoothly they could walk, whereas around a third of the participants expressed that their movement pattern only was affected a little. Some described that they no longer had the same ability to control the affected lower limb.


*Something has happened to the weakest leg, sometimes it’s just as if it doesn´t want to follow. The foot, or yes, the lower leg… Has no control over it. I didn´t have that problem before, but I do now…*
 [P10]

One participant described that it was extra difficult to control the lower limb when walking outdoors. He needed completely flat ground and said that if something small was in his way, the fall risk increased substantially. 


*When I’m outside walking it’s just as if my right leg [jams itself] in a strange way right at the moment when I put it down and my left leg leaves the ground. I need the ground to be completely flat. Just something small in the way is enough.*
 [P2]

#### 3.2.12. Concentration When Walking (Item 12) 

Most of the participants (71%) reported that their disability following LEoP had forced them to concentrate, moderately or quite a bit, whilst walking. They described that it was quite energy-intensive to have to concentrate on the walking, for example in hilly terrain or in situations where there were a lot of people. 


*It takes a lot of energy to concentrate on how to walk and it´s of course affected if there are a lot of people in the store.*
 [P5]

Having to constantly concentrate and being tense when walking, for example in the forest, meant that some had stopped doing such activities. The daylight also affected how much the participants needed to concentrate on walking. One strategy to keep balance and to avoid falling was to concentrate on every step, which was perceived as exhausting, but also to take it easy and to look where they put their feet. Doing so increased safety when they walked. 


*I look down more, look where I put my feet. I never had to do that before. I do it for safety.*
 [P4]

## 4. Discussion

To the best of our knowledge, this is the first study that has explored, using a mixed-methods design, how self-perceived walking limitations influence daily life in persons with LEoP. By using both a rating scale and individual interviews, a deeper understanding of the participants’ walking limitations was obtained. The participants perceived that running was most limited, whereas walking indoors without using support was least limited. A majority were also moderately to extremely limited in activities related to standing or walking. The limitations intruded on many everyday activities, both indoors and outdoors, which restricted social participation. To increase safety when walking and to reduce the fall risk, various strategies were adopted such as using aids, walking more carefully, and avoiding risky activities.

Many participants rated that running was extremely limited, and more than 70% perceived that their ability to walk, to maintain balance, to walk longer distances and to walk smoothly were moderately to extremely limited. Additionally, many had to slow down their walking speed, to concentrate on walking, to use support in standing and when walking outdoors and climbing stairs as a consequence of their LEoP. Our findings are in agreement with previous studies of persons with LEoP [5,6,15], that have used Walk-12 as a rating scale to capture self-perceived walking limitations in daily life. Similar limitations have also been reported in persons with stroke [13], MS, and PD [20], indicating that many activities related to standing and walking are challenging for persons with different neurological conditions. To use Walk-12 as a screening tool of walking limitations in daily life, can provide clinicians a quick understanding of the problems that people may perceive.

The deductive analysis revealed that limited balance when standing or walking, limited ability to walk, and need of support when walking indoors were aspects that were most commonly mentioned during the interviews. The participants said that ageing with LEoP often meant reduced balance and increased muscle weakness in the lower limbs, but also increased pain and fatigue which gradually decreased their ability to stand up and to walk. To manage their personal hygiene was described as rather difficult, but also to perform household activities while standing. Additionally, walking in different environments outdoors, for example, in crowded areas or on uneven ground, as well as climbing stairs, could be very challenging for some. All these difficulties in daily life restricted their ability to participate in social events, such as meeting with friends and attending sporting events. A previous study among 325 persons with LEoP [6] has shown that LEoP-related impairments, such as muscle weakness, muscle fatigue, muscle and/or joint pain during physical activity and general fatigue, are associated with walking limitations to various degrees. Another study in persons ageing with LEoP [21] has described that the impairments can also increase in the less affected leg. The decline in functioning in both lower limbs makes the persons even more vulnerable, as it further reduces their walking ability and possibility to be physically active [22] and participate in meaningful everyday activities [23]. This is important to consider in the rehabilitation of people with LEoP.

Furthermore, the participants in our study described that ageing with LEoP meant that they now had to rely more on aids in order to decrease pain, fatigue and fall risk. Many different aids were used to increase safety, such as a stool in the kitchen, and a shower board in the bathroom. They also used a variety of walking aids, for example dictus bands, ankle–foot orthoses, canes, rollators and Nordic Walkers. Some even used a shopping cart as a support in grocery stores. Previous studies in LEoP have shown that different types of walking aids [24] and ankle–foot orthosis (AFO) [25,26] could be helpful to increase safety and balance during walking. An AFO may improve gait speed, outdoor walking, and reduce perceived exertion, but also improve the feeling of increased stability when walking [25]. People ageing with LEoP seem to have a more positive attitude towards mobility aids when they are growing older [23]. One explanation may be that they, over time, have accepted the decline in functioning and have learned to live with their disability. As other adults of similar age also use more mobility aids, it is less of a stigma to use aids and easier to blend in when using them. 

Other aspects that our participants described during the interviews were that they needed completely flat grounds in order not to stumble and fall. Inaccessible environments, such as slopes and stairs, as well as slippery surfaces, meant that it could be difficult going out and running errands. Sometimes during the winter, they had to wait to go out until the ground was sanded, due to anxiety and fear of falling. Additionally, not being able to manage certain activities, for example walk in the forest or participate in social events if there were stairs in the buildings, sometimes meant that they gave up meaningful activities. It has been reported that persons with LEoP [27], but also persons with PD [28], often avoid activities related to walking, such as going out when it is slippery, walking longer distances, and attending social events because of the fear of falling. Even if it can sometimes be beneficial to avoid activities that pose a high risk of falling, it may be emotionally difficult and frustrating not being able to perform meaningful activities anymore [24,27]. 

To be able to cope with challenges in daily life related to standing and walking, our participants had developed several strategies. In addition to the fact that many used more aids to support themselves indoors as well as outdoors, they had also learned not to walk when it was dark and/or slippery outside. To focus more on walking, i.e., to think on every step and where to put the feet and to walk with slower pace, was important to increase safety and reduce the fall risk. It was also a common strategy to pause and to stand still to regain balance when standing up, for example from a chair. Over the years, they had also learned to listen more to the body and to allow themselves to rest when they got tired. These strategies are in line with previous studies [23,29], and indicate that persons ageing with LEoP have come to terms with their situation and adapted to their disability over time.

### 4.1. Methodological Considerations and Clinical Implications

Walk-12 is shown to have good measurement properties [17] and it covers several aspects of walking limitations in daily life that persons with LEoP may perceive. However, by using face-to-face interviews, we have expanded and deepened our understanding of how walking limitations influence participants’ everyday life. The limitations in standing and walking, which increased the fall risk and caused emotional reactions of worry and fear of falling, underline the importance of understanding the impact on everyday activities and social participation. It has been shown that the decline in walking (using objective measures) develops slowly over time in people with LEoP [30,31]. Hence, by providing various targeted rehabilitation interventions, such as reducing LEoP-related impairments, adapting the home environment, prescribing appropriate aids and discussing strategies to achieve a good balance between rest and physical activity, walking ability may be sufficiently maintained. Being able to walk has been shown to be significant for people with PD [32], but a major contributing factor to fear of falling [33]. Thus, future studies need to investigate how walking ability in daily life changes over time in persons ageing with LEoP. Additionally, more studies that evaluate the efficacy of various rehabilitation interventions aimed at improving walking ability are warranted.

### 4.2. Strength and Limitations

A strength of this study was that the participants had various degrees of walking limitation following LEoP, and that men and women were equally represented. Fourteen persons with LEoP were included, and they provided rich and relevant data. As no further information was added from the last interviews, we decided not to recruit any more participants. The mixed-methods design enabled us to triangulate the data [18], leading to a broader and a more in-depth understanding of how participants’ walking limitations influenced daily life and strategies they used to manage daily activities. The data from the Walk-12 were analysed after the interviews to avoid being influenced by the results. During the process of analysing and coding the data, the authors worked individually. However, in order to maintain reflexivity, the authors had continuous discussions during the analysis to stay aware of our preunderstanding. To add transparency and trustworthiness to our findings, all steps of the data collection and the analysis were described and we added the participants’ identity number after each quotation [34]. The study has also some limitations. All participants were of Swedish origin and older than 60 years. Therefore, transferability to persons of other origins and younger persons with LEoP is limited and needs to be further studied.

## 5. Conclusions

This study has shown that persons with LEoP experience walking limitations to a great extent. Running was perceived as most limited, whereas walking indoors without using support was perceived as least limited. Many were moderately to extremely limited in standing or walking, in walking speed and distance, which affected concentration and effort, as well as gait quality aspects. The walking limitations negatively affected their ability to perform many daily activities, led to a fear of falling, and restricted social participation. To increase safety when walking and to reduce the fall risk, various strategies were used. The Walk-12 could be a valuable screening tool to obtain knowledge on how many walking limitations persons with LEoP perceive, whereas the interviews broadened our understanding of the influence on everyday life. By using a mixed-methods design, a deeper understanding of the consequences of walking limitations was achieved. The knowledge could be important for clinicians when planning rehabilitation interventions aiming at improving walking ability in people with LEoP.

## Figures and Tables

**Table 1 ijerph-19-08157-t001:** Demographics and characteristics of the 14 participants with late effects of polio.

Age	
Mean years (SD), range	70 (5), 61–78
**Age at acute polio**	
Mean years (SD), range	4 (4), 1–12
**Duration of new LEoP-related impairments**	
Mean years (SD), range	26 (9), 9–43
**Self-perceived degree of LEoP**	
Mild/moderate/severe (*n*)	3/6/5
**Use of mobility aids**	
Indoors or outdoors (*n*)	11
**Living situation**	
Living with a partner/alone (*n*)	11/3
**Resident**	
Living in a house/apartment (*n*)	8/6
**Ground around the residence**	
Even/uneven (*n*)	12/2
**Fall frequency the past year**	
0 time/1–3 times/4–6 times	2/7/5

SD = standard deviation. LEoP = late effects of polio.

**Table 2 ijerph-19-08157-t002:** Self-perceived walking limitations (Walk-12) reported by the 14 participants with late effects of polio.

In the Past 2 Weeks How Much Has Your Post-Polio…	Not at All *n* (%)	A Little *n* (%)	Moderately *n* (%)	Quite a Bit *n* (%)	Extremely *n* (%)
1. Limited your ability to walk?	3 (21)	1 (7)	4 (29)	4 (29)	2 (14)
2. Limited your ability to run?	0	2 (14)	1 (7)	2 (14)	9 (64)
3. Limited your ability to climb up or down stairs	1 (7)	4 (29)	2 (14)	4 (29)	3 (21)
4. Made standing when doing things more difficult?	3 (21)	2 (14)	4 (29)	4 (29)	1 (7)
5. Limited your balance when standing or walking?	1 (7)	3 (21)	5 (36)	4 (29)	1 (7)
6. Limited how far you are able to walk?	0	4 (29)	0	7 (50)	3 (21)
7. Increased the effort needed for you to walk?	0	4 (29)	2 (14)	6 (43)	2 (14)
8. Made it necessary for you to use support when walking indoors, e.g., holding on to furniture, using a stick, etc.?	5 (36)	1 (7)	3 (21)	4 (29)	1 (7)
9. Made it necessary for you to use support when walking outdoors, e.g., using a stick or frame, etc.?	4 (29)	1 (7)	1 (7)	5 (36)	3 (21)
10. Slowed down your walking?	1 (7)	3 (21)	0	8 (57)	2 (14)
11. Affected how smoothly you walk?	0	4 (29)	0	7 (50)	3 (21)
12. Made you concentrate on your walking?	0	4 (29)	3 (21)	7 (50)	0

Response options: Not at all = 1; A little = 2; Moderately = 3; Quite a bit = 4; Extremely = 5. *n* (%) = number (percent).

## Data Availability

All data were archived according to the Swedish Act concerning the Ethical Review of Research Involving Humans and are available upon reasonable request.

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
