# Peer review of "The Influence of Walking Limitations on Daily Life: A Mixed-Methods Study of 14 Persons with Late Effects of Polio"

_ijerph, 2022, doi:10.3390/ijerph19138157_

Round 1

Reviewer 1 Report

Dear authors,

Congratulations for your work.

General comments

The authors tried to demonstrated the effects of self-perceived walking in persons with late effects of polio have in the daily life. The study  showed that these person are affected in the daily life and have the fear of falling which restrict their social participation.

Here are my considerations:

Title: Add study case to your title since your sample is to small

Abstract: Line 14-15- rewrite the methods. They are not clear

Introduction: Line 53: Add reference that state the same

Methods: Add exclusion criteria; How do you know 7 men and 7 women are enough to do your study? Do you use G*Power?

Discussion: What are the strengths of your work by using this two methods? You have to discuss this novelty.

Your study doesn´t have no limitations? The sample size is one of them.

Conclusions: Line 433-436 – you have to strength this conclusion by adding two paragraphs in the discussion about it.

Reviewer 2 Report

The submitted work touches on a perennially important issue both medically and socially. Gait disturbances are important symptoms and consequences in numerous neurological diseases of central and peripheral neuronal origin, diseases of the musculoskeletal system or congenital defects. This enumeration indicates the constant topicality of the subject of the study. Thus, it was with all the more interest that I accepted the authors' proposal to use a mixed research method, combining an already recognized tool of quantitative analysis (Walk-12) with a tool based on a strictly qualitative phenomenological approach in humanistic psychology. The idea of a more holistic approach to patients and the study of their problems came to my mind. I would be very interested in a comparative study, based on a large group of subjects, in which the usefulness of the method would be verified against other already existing methods.

Unfortunately, the submitted paper presents the results of a study of only 14 patients, 7 women and 7 men. I am aware of some difficulty in reaching a larger number of polio patients, but any conclusion based on such a small cohort is simply impossible. Rather, the results obtained should be seen as a guide to the planning of a larger study in which one could go beyond patients with a history of polio to form a study group with, for example, gait disorders due to motoneuron dysfunction in genere. In my opinion, the paper is worthy of publication, but its primary aim should be to propose a new research method, and the results of its application in a group of 14 people can only serve as an example of its application, but without any medical or social conclusions. Based on the above, I propose to change the title of the paper and to reword it in order to place the main emphasis in the place I have indicated.

Reviewer 3 Report

Dear authors.

The paper is original and gives a different point of view about the impact of postpolio syndrome on patients´ activities.

Maybe is more a sociological research than a medical research, but is done in deep and clearly shows patients´ dissabilities

In order to point some aspects that can be improved (but more as a suggestion than a necessity) I will say:

* point 2.2 describes the participants. For me will be better to be readen in results than in Materials and method. Table 1 (the most important in research to give readers an image of population) is usually a results table.

* point 3.2. throughout the presentation of the results we can read many indeterminate values ("more than half", "almost all", "a majority"...). It will be better if readers can know exactly how many mean these terms. Be more concise.

Round 2

Reviewer 1 Report

Dear authors,

you  have attended the majority of my suggestions and improved the article.

I have no more suggestions.

Reviewer 2 Report

The authors have submitted a revised version of the article. The added extracts respond in large part to earlier comments. Due to the fact that it is not possible to extend the study to a larger group of patients, I believe that the changes made are sufficient to consider the paper ready for publication